# Current Immunotherapeutic Strategies for the Treatment of Glioblastoma

**DOI:** 10.3390/cancers13184548

**Published:** 2021-09-10

**Authors:** Mark Dapash, Brandyn Castro, David Hou, Catalina Lee-Chang

**Affiliations:** 1Pritzker School of Medicine, University of Chicago, Chicago, IL 60637, USA; mark.dapash@uchospitals.edu; 2Department of Neurological Surgery, Feinberg School of Medicine, Northwestern University, Chicago, IL 60611, USA; brandyn.castro@uchospitals.edu (B.C.); david.hou@northwestern.edu (D.H.); 3Department of Neurosurgery, University of Chicago, Chicago, IL 60637, USA; 4Northwestern Medicine Malnati Brain Tumor Institute, Lurie Comprehensive Cancer Center, Feinberg School of Medicine, Northwestern University, Chicago, IL 60611, USA

**Keywords:** glioblastoma, immunotherapy, glioblastoma immunotherapy, checkpoint inhibitors, vaccine, CAR-T

## Abstract

**Simple Summary:**

Glioblastoma is the most common primary brain tumor in adults, and its aggressive nature yields a poor prognosis despite current treatment strategies. The aim of our literature review is to discuss various immunotherapeutic strategies currently being investigated in the treatment of glioblastoma. Checkpoint inhibitors, various vaccination strategies, and CAR T-cell therapies serve as some of the most investigated immunotherapeutic strategies. However, all strategies face various limitations, such as the low relative mutational burden, the immunosuppressive tumor microenvironment, and genetic heterogeneity, which serve as the current challenges.

**Abstract:**

Glioblastoma (GBM) is a lethal primary brain tumor. Despite extensive effort in basic, translational, and clinical research, the treatment outcomes for patients with GBM are virtually unchanged over the past 15 years. GBM is one of the most immunologically “cold” tumors, in which cytotoxic T-cell infiltration is minimal, and myeloid infiltration predominates. This is due to the profound immunosuppressive nature of GBM, a tumor microenvironment that is metabolically challenging for immune cells, and the low mutational burden of GBMs. Together, these GBM characteristics contribute to the poor results obtained from immunotherapy. However, as indicated by an ongoing and expanding number of clinical trials, and despite the mostly disappointing results to date, immunotherapy remains a conceptually attractive approach for treating GBM. Checkpoint inhibitors, various vaccination strategies, and CAR T-cell therapy serve as some of the most investigated immunotherapeutic strategies. This review article aims to provide a general overview of the current state of glioblastoma immunotherapy. Information was compiled through a literature search conducted on PubMed and clinical trials between 1961 to 2021.

## 1. Introduction

Glioblastoma (GBM) is the most malignant of the glial tumors (grade IV) and represents more than half of all primary brain tumors in the United States, with an annual prevalence of roughly 3.19 per 100,000 people [1,2]. GBMs provide a unique challenge due to their highly invasive nature, leading to a nearly 100% recurrence rate even in cases of gross total resection based on radiographic criteria [3]. Moreover, a poor response to existing treatment is usually seen due to the inherent intra-tumor heterogeneity [4,5,6]. The median survival of patients with GBM is approximately 14 months, even with aggressive treatment regimens [2,7]. 

The Stupp protocol remains the standard of care for GBM, with virtually no improvements since it was developed in 2005. The Stupp protocol entails the administration of temozolomide (TMZ) (75 mg/m^2^) with concomitant radiotherapy (RT) (60 Gy) for 42 days followed by six cycles of adjuvant TMZ (150 to 200 mg/m^2^) administered for five days during each 28-day cycle [8]. In 2017, tumor-treating fields and maintenance temozolomide were shown to increase progression-free survival and overall survival in patients when compared to maintenance temozolomide alone [9]. Still, there is much necessity for novel treatment strategies and novel therapeutic targets. The FDA’s approval of interferon-alpha 2 (IFN-α2) for the treatment of hairy cell leukemia in 1986 marked a major transitional point in cancer-directed immunotherapy [10]. Since then, there have been many pre-clinical and clinical developments in the treatment of solid and hematologic malignancies [11,12]; immunotherapy has emerged as a promising approach for GBM therapy.

No longer thought to be an immune privilege, the dynamic immune response and active immune-surveillance of the central nervous system (CNS) have been highlighted in numerous studies [13,14,15,16]. The roles that microglia, macrophages, and dendritic cells play as potent antigen presenter cells (APCs) in the CNS have been further established [17,18,19,20]. In 2015, Louveau et al. provided experimental evidence for the meningeal lymphatic vessels (MLVs) and their role in facilitating the transport of APCs to the nasal and deep cervical lymph nodes [21]. These APCs go on to subsequently prime B- and T-lymphocytes. These studies, along with emerging evidence from ongoing studies, help highlight the potential of immunotherapy in GBM treatment. This literature review covers the established and emerging immunotherapeutic treatment modalities and discusses unique challenges faced when targeting GBMs. Table 1 summarizes our findings.

## 2. Results

### 2.1. Checkpoint Blockade

Checkpoint inhibitors are tumor-directed monoclonal antibodies that have established therapeutic efficacy in the treatment of various solid tumors such as lung, head-and-neck, and renal cancers in addition to melanoma [22,23,24,25]. These antibodies specifically target immune checkpoint molecules (ICs), which normally function to attenuate T-cell function (Figure 1). Cytotoxic T-lymphocyte-associated protein 4 (CTLA-4) and programmed cell death protein 1 (PD-1) are two of the most researched ICs. Inhibiting them has been shown to significantly augment the antitumor response [26,27]. CheckMate 143 served as the first randomized phase I clinical trial that targeted the PD pathway in recurrent GBM. In this study, nivolumab (a PD-1 inhibitor), in addition to ipilimumab (a CTLA-4 inhibitor), were evaluated for safety and efficacy. All 40 patients in this study received surgical resection, radiation, and temozolomide before being divided into three treatment arms. Objective response rates (ORR) of 11% and 10% were observed in the NIVO3 and NIVO1+IPI3 treatment arms, respectively. The phase I trial highlighted that nivolumab monotherapy was better tolerated than nivolumab plus ipilimumab. Additionally, the tolerability of the combination therapy was seen to be influenced by ipilimumab dosage [28]. The subsequent phase III clinical trial randomized 369 recurrent GBM patients to receive either nivolumab or bevacizumab, an antiangiogenic antibody that targets VEGF-A, which received accelerated FDA approval in recurrent GBM in 2009 [29,30]. Results from this study did not demonstrate an improved OS in patients with recurrent GBM when compared to bevacizumab monotherapy. When investigating the overall response rate (ORR), which describes the number of patients who have experienced a complete or partial response to therapy, it was noted to be lower with nivolumab than with bevacizumab [31]. 

Durvalumab (durva) is a human IgG1 monoclonal antibody that specifically targets PD-L1. Durva has already been approved in the treatment of non-small cell lung cancer in addition to bladder cancer in select patients [32,33]. A phase II trial is currently investigating the efficacy and safety of durva in the treatment of newly diagnosed GBM. This trial served as the first study of anti-PD-L1 antibodies in the treatment of newly diagnosed GBM. There are five GBM cohorts; the data was published for cohort A. Cohort A evaluated durva, in addition to the radiotherapy (60 Grays over 30 fractions), followed by durva monotherapy in 40 patients with newly diagnosed GBM. No significant differences in overall survival were noted with this treatment regimen. The treatment was well tolerated when combined with radiotherapy and the adverse events (grade ≥ 3), as defined by the Common Terminology Criteria for Adverse Events (CTCAE), occurred in 35% of patients (14)—with the most common adverse event being the asymptomatic increase in lipase (six patients) and amylase (two patients). Another ongoing phase II study aims to determine the safety and efficacy of tremelimumab (CTLA-4 inhibitor) and durva as monotherapies or in combination as adjuvant therapy for recurrent GBM [34]. The combination of varlilumab, an agonist anti-CD27 monoclonal antibody, and nivolumab has been investigated for refractory solid tumors [35] in a phase I/II trial that monitored for adverse effects, dose-limiting toxicities, and laboratory abnormalities 100 days from the last study drug dose in 36 patients. CD27 is a known costimulatory molecule that is able to stimulate T-cells to proliferate, differentiate, and increase their effector response. The solid tumors investigated were ovarian carcinoma, colorectal cancer, melanoma, and squamous cell carcinoma of the head and neck. There were no unexpected toxicities, and there was encouraging evidence of antitumor activity in subsets of patients prompting further investigation. The subsequent trial will investigate the efficacy of the combination varlilumab and nivolumab at different dosages as measured by the overall survival of 12 months in GBM, along with the other previously investigated solid tumors.

Aside from the Checkmate 143 phase III trial, ORR has been demonstrated in two other checkpoint inhibitor studies [36,37]. A single practice case series demonstrated an ORR of 31% in a total of 20 patients with recurrent GBM who were treated with ipilimumab plus bevacizumab [36]. Although two patients could not complete the treatment regimen due to grade 2 adverse events, the treatment combination was well-tolerated overall [36]. A phase I study of atezoliumab, an antibody targeting programmed cell death-ligand 1 (PD-L1), demonstrated an ORR of 6% in 16 patients with recurrent GBM [37]. Despite the limited therapeutic efficacy of the checkpoint blockade in recurrent GBM, there has been enthusiasm in evaluating their efficacy in the neoadjuvant setting. In 2019, a single-arm phase II clinical trial analyzed pre- and post-surgical administration of nivolumab for 30 GBM patients, 27 with recurrent disease and 3 with newly diagnosed tumors. Despite the promising results of higher immune cell infiltration and T-cell receptor clonal diversity among tumor-infiltrating T lymphocytes, no clear clinical benefit was shown following salvage surgery for recurrent cases [38]. Another multi-institutional randomized controlled trial evaluating the PD-1 inhibitor pembrolizumab was conducted by the Ivy Foundation Early Phase Clinical Trials Consortium in 35 patients with recurrent GBM. In this study, patients who received neoadjuvant and adjuvant pembrolizumab had a median OS of 417 days versus a median OS of 228.5 days in patients receiving adjuvant pembrolizumab only [39]. Although limited by small sample size, the early results supported the expansion of the study and the pursuit of future clinical trials.

Ongoing trials continue to investigate CTLA-4, PD-1, and other potential checkpoint inhibitors such as indoleamine 2,3-dioxygenase (IDO). IDO is an endogenous enzyme that plays an important role in the regulation of the immune system functioning to augment suppressor activity of regulatory T-cells and consequently inhibit CD8+ T-cells [40,41]. Studies have shown that 50 to 90 percent of GBMs express IDO, which is correlated with poor prognosis [42,43]. In vitro, IDO inhibitors have been shown to slow tumor growth through the improvement of anti-tumor T-cell responses [44]. A phase Ib/II trial is currently ongoing to evaluate the IDO inhibitor indoximod in newly diagnosed GBM patients [45]. Their early work in mouse models has highlighted a synergistic effect of indoximod when used in conjunction with temozolomide and radiation. The primary endpoint of the phase II trial will be six months PFS. Epacadostat, another IDO inhibitor, is being investigated in an ongoing phase I/II trial. This trial aims to identify the safety and feasibility when administered with nivolumab in subjects with advanced solid tumors and lymphomas [46]. Phase II will include expansion cohorts in 7 tumor types, including GBM.

### 2.2. Peptide-Based Vaccines 

Vaccination is another method in immunotherapy in which a tumor-specific response is provoked with the use of a foreign antigen. Vaccines can be either cell or peptide-based, and both types have been successfully investigated in clinical trials. Rindopepimut is a peptide-based vaccine targeting the EGFR deletion mutation in variant III of the epidermal growth factor receptor (EGFRvIII), commonly seen in GBM. It was first studied in the single-group phase II trial ACTIVATE, where 100 patients with EGFRvIII positive GBM were given rindopepimut alone, following gross total resection and lack of progression, following standard treatment of radiotherapy with concomitant temozolomide [47]. This was subsequently followed up with two additional phase II trials, ACT II and ACT III. Both trials focused on rindopepimut with adjuvant temozolomide following gross total resection and no evidence of progression after standard treatment. The results in these three studies were promising as they resulted in progression-free survival of roughly 15 months from the time of diagnosis and overall survival of 24 months when compared to the cohort which received the standard treatment [48,49]. ACT IV was a randomized, double-blind phase III trial that investigated whether rindopepimut plus standard treatment would improve overall survival in patients with minimal residual disease (MRD). MRD was defined as having an enhancing tumor that is less than 2 cm^2^ following surgery and chemoradiation. Around 370 patients were randomized into each arm, and at the final analysis, there was no significant difference in overall survival for patients with MRD. This was unexpected due to the multiple, independent phase II trials that seemed to suggest benefits from the use of rindopepimut [50].

Another peptide-based vaccine being investigated in the treatment of GBM utilized the Wilm’s tumor (WT1) gene. The WT1 gene plays an important role in controlling cell proliferation and apoptosis. The WT1 gene is a widely recognized oncogene that is overexpressed in various solid malignancies [51,52,53]. It is diffusely present in various grades of glioma, including GBM. Furthermore, it is associated with negative prognostic factors such as IDH-wild type expression and older age [54]. Izumoto et al. conducted a phase II study in which they enrolled 21 patients with WT1/HLA-A*2402-positive recurrent GBM that had failed standard therapy. Although no patient showed a complete response, two patients showed partial responses. The median PFS was 20 weeks, with a 6-month PFS rate of 33.3% [55]. Hashimoto et al. investigated the safety profile of a WT1 peptide-based vaccine and temozolomide for the treatment of newly diagnosed GBMs. This study also demonstrated an acceptable safety profile, and from the time of diagnosis, the seven patients enrolled had a progression-free survival ranging from 5.2 to 49.1 months [56].

The IDH1 peptide and Survivin are the other two molecules being investigated in peptide-based vaccines. SurVaxM(SVN53-67/M57-KLH) is a peptide vaccine synthesized to target survivin, a protein that aids in the inhibition of apoptosis. Survivin is expressed on GBM tumors and is associated with a poor prognosis [57]. Early studies using murine GL261 tumor cells and human glioma cells ex vivo showed that SurVaxM could incite an anti-tumor immune response [58,59]. There is a current phase II clinical trial (NCT02455557) evaluating the clinical effects of the vaccine as well as its immunogenicity [60]. Similarly, based on promising results in murine models, the German National Cancer Center in 2015 launched the NAO-16 phase I trial (NCT02454634) of the IDH1 peptide vaccine. This vaccine is engineered to specifically target the IDH1R132H mutation [61]. Additionally, Duke University also has an ongoing clinical trial (NCT02193347) investigating the use of an IDH1 peptide vaccine in the treatment of recurrent grade II gliomas [62]. The German National Cancer Center’s trial focuses on grade III and IV gliomas versus Duke’s trial, which is also using this peptide-based vaccine but in recurrent grade II gliomas.

In the past decade, targeting multiple tumor-related peptides in a single vaccine has also become another approach in treating GBM. In 2012, Dutoit et al. used liquid chromatography-mass spectrometry to identify over 3000 peptides specifically expressed in HLA-A*02-positive GBM. From this pool of peptides, 11 (baculoviral inhibitor of apoptosis protein repeat-containing 5, brevican, chondroitin sulfate proteoglycan 4, fatty acid-binding protein 7- brain, insulin-like growth factor 2 mRNA binding protein 3, neuroligin 4- X-linked, neuronal cell adhesion molecule, Met proto-oncogene, protein tyrosine phosphatase -receptor-type, tenascin C, Z polypeptide 1, and hepatitis B virus core antigen) were selected and used in the creation of the IMA950 [63]. Forty-five patients were enrolled in the first phase I/II trail of IMA950 plus GM-CSF for newly diagnosed GBM following surgical resection and standard therapy. At six months PFS was 74%, and at nine months PFS was 31% [64]. A second phase I trial investigating IMA-950 in combination with cyclophosphamide, GM-CSF, and imiquimod was terminated early due to poor accrual [65]. Clinical trials investigating other multi-peptide vaccines are also underway. SL701 is another newly developed multi-peptide vaccine that targets three peptides that are over overexpressed in gliomas, ephrin A2, survivin, and IL-13 receptor α-2 [66]. 

Heat shock proteins (HSP) can also be utilized in peptide-based vaccines. Heat shock proteins function to control the degradation of misfolded proteins along with modulation protein aggregation in response to cellular stress. HSP70 and HSP96 are two heat shock proteins that have been found to elicit a pro-inflammatory response to GBM tumor-associated antigens (TAAs) [67]. Clinical trials up to this point have focused mainly on HSP96 and the used HSP–peptide complex-96 (HSPPC-96) vaccine. A phase I trial investigating an HSPPC-96 vaccine demonstrated specific immune responses in 11 of the 12 patients enrolled [68]. A follow-up single-arm phase II trial demonstrated a median progression-free survival of 19.1 weeks and OS of 42.6 weeks following gross total resection in patients with recurrent GBM [69]. Phase II trials of HSPPC-96 yielded promising results prompting sponsorship by the Alliance for Clinical Trials in Oncology to conduct a multi-institutional trial (NCT01814813) of HSPPC-96 for the treatment of recurrent GBM. This study aims to provide evidence that the adjuvant treatment with the HSPPC-96 vaccine can prolong overall survival. This is a three-armed study that includes HSPPC-96 with the administration of bevacizumab at tumor progression, HSPPC-96 with concomitant bevacizumab, and bevacizumab alone. The primary outcome being measured is OS, in addition to the secondary outcomes evaluating PFS, safety, and tolerability of the combined therapy [67].

Neoantigens are proteins derived from tumor-specific mutations that arise in protein coding and are also attractive candidates for peptide vaccine development. These neoantigens have been shown to generate a robust immune response and can function to facilitate tumor rejection [70,71,72]. GAPVAC-101 was a phase I trial that investigated highly individualized vaccinations with both neoepitopes and unmutated antigens when administered with the standard of care [73]. The primary aim of the study was to evaluate the immunologic response, efficacy, and safety of this treatment modality in 15 patients with newly diagnosed GBMs. The vaccination schedule consisted of the APVAC1, which targeted human leukocyte antigen (HLA)-A*02:01 or HLA-A*24:02, followed by treatment with APVAC2, which selectively targeted neoepitopes. Through analyses of immunopeptidomes and transcriptomes from each patient’s tumor, mutations that resulted in neoepitopes were identified. The APVAC1 vaccines were able to elicit a sustained CD8 T-cell response, while APVAC2 was able to elicit a primarily CD4 T-cell response, particularly the Th1 type. In these patients, reactivity to APVAC1 was seen in 12 of 13 patients, and responses to APVAC2 being seen in 8 of 10 patients who were able to be evaluated. Reactivity was defined by CD4^+^ response utilizing an IFNγ enzyme-linked assay. In addition to the strong immunologic response, the vaccines proved to be well tolerated. Consequently, a phase Ib study was conducted with eight patients that were immunized with personalized neoantigen vaccines [74]. Two of eight patients generated a primarily CD4+ T-cell response to several of the vaccine antigens; however, there was no significant difference in median PFS or OS when compared to historical controls. 

### 2.3. Cell-Based Vaccines

Dendritic cells (DCs) are antigen-presenting cells (APCs) that serve as a vital connection between innate and adaptive immunity. They are found in nearly every tissue and are known to be potent stimulators of T- and B-cells [75]. Consequently, DCs have been an area of interest when developing cellular vaccination strategies against tumors. Previous studies have been able to highlight the effectiveness of DC vaccination (DCV) for gliomas in preclinical models as well as early-stage clinical trials [76,77]. Wheeler et al. conducted a study in which 32 patients were given a DC-based vaccine, loaded with autologous tumor lysate, and vaccine responders and non-responders were compared. Of these patients, 53% exhibited enhanced IFN-y responses, and responders exhibited longer time to progression and time to survival [78]. A pivotal phase I/II clinical trial by Yamanaka et al. enrolled 18 patients with recurrent GBM. This study examined the use of a DCV that was generated with IL-4, GM-CSF, and pulsed together with autologous tumor lysate with or without OK-432 (agent derived from killed *Streptococcus pyogenes*). This study also included a subset of patients who received intratumoral administration of the DCV via an Ommaya reservoir in addition to the standard intradermal injections. This study demonstrated the safety of DCV with no observed adverse events or radiographic evidence of autoimmune reactions. In addition to a statistically significant median OS seen in the treatment group (480 days vs. 400 days), there was an increased T-cell reactivity to tumor lysate observed post-vaccination [79]. Fadul et al. also investigated the use of a DC-based vaccine in which the DCs were loaded with autologous tumor lysate. In this study, after TMZ and radiation therapy, 10 patients received the vaccine. In these patients, the overall survival was 28 months, with a median PFS of 9.5 months [80]. In 2013, Vik-Mo et al. investigated the use of a DC-based vaccine for targeting GBM stem cells and found that all seven patients in the study exhibited an immune response, and consequently, a PFS that was 2.9 times longer than their matched controls [81]. 

ICT-107 is a patient-specific DC-based immunotherapy for newly diagnosed GBM patients. This immunotherapy utilizes TAAs present on GBM cells to create six synthetic peptides that are then pulsed with the patient’s DCs. In 2013, ICT-107 was studied in a double-blind, placebo-controlled phase II trial to evaluate its safety and efficacy when administered in conjunction with the Stupp protocol for patients with newly diagnosed GBM [82]. This trial highlighted that prolonged OS correlated with the expression of four ICT-107 targeted-antigens in pre-vaccinated tumors. Despite this encouraging result, the OS benefit was not shown in the later phase II trial when compared to the standard of care [83]. In 2015, a randomized, double-blind phase III trial was conducted to compare the standard of care to ICT-107. The study was suspended before reaching its primary outcome due to insufficient financial resources [84]. Another DC vaccine study, called GlioVax, is currently ongoing and is a phase II randomized-controlled clinical trial (NCT03395587) seeking to confirm promising results of earlier, smaller phase I/II trials [85]. This study aims to recruit 136 patients with newly diagnosed, IDH-wildtype GBMs, and assign them to the standard of care of radiotherapy with temozolomide chemotherapy or the standard of care plus DCV. A phase III trial analyzing DCVax^®^-L, an autologous tumor lysate-pulsed DCV, showed great potential for this vaccination strategy when combined with standard therapy for a newly diagnosed GBM [86]. Blinded interim survival data showed the median OS for the intent-to-treat (ITT) population was 23.1 months from surgery. This was an improvement from the median OS for the standard of care. Interestingly, patients with MGMT methylation had a median OS of 34.7 months from surgery with a 3-year survival of 46.4%. Furthermore, there was a population of extended survivors (*n* = 100) with a median OS of 40.5 months, which will be further analyzed. However, the study did not provide the IDH status of these patients. The adverse events with DCVax^®^-L were comparable to standard therapy alone. Final results from further analyses and follow-up have not yet been published. 

Human umbilical vein endothelial cell (HUVEC) vaccines are less commonly investigated for the treatment of GBM. It is believed that HUVEC antigens elicit cellular and humoral immune responses that are antiangiogenic; thus, inhibiting tumor growth [87,88,89]. This is particularly important because bevacizumab has only shown limited clinical benefits in recurrent GBMs [90]. Thus far, clinical trials investigating HUVEC for recurrent GBM have shown to be well tolerated and have yielded encouraging early results [90,91]. Other techniques for tumor cell vaccine delivery involve formalin fixation of the tumor cells before injection of the vaccine. It has been demonstrated that fixation with formalin allows for better preservation of the tissue, which allows for the most robust immune response against the tumor cells [92]. The safety and efficacy of autologous formalin-fixed tumor vaccines (AFTVs) were tested in two clinical trials examining its use with only fractionated radiotherapy and with chemoradiation in patients with newly diagnosed GBM [92,93]. Both trials were able to demonstrate a tolerable safety profile, and both trials yielded a median OS of greater than 19 months. These encouraging results prompted a prospective placebo-controlled phase IIb/III trial evaluating AFTV therapy in combination with standard chemoradiotherapy [94]. Although preliminary results confirmed the safety of AFTV therapy, this trial was not able to find a statistically significant difference in median PFS between the two experimental arms.

### 2.4. Oncolytic Viruses

Several studies have been able to establish that Cytomegalovirus (CMV) proteins and are expressed in over 90% of GBM tumors [95,96,97]. Furthermore, these proteins are not expressed in the surrounding normal brain tissue [95,96,98,99]. In 2014, Nair et al. demonstrated that T-cells specifically targeting CMV were able to effectively target and kill GBM tumor cells that express the pp65 antigen [100]. A subsequent randomized pilot trial was able to demonstrate the efficacy of CMV pp65-specific dendritic cells (pp65-DCs) when combined with vaccine site pre-conditioning using tetanus-diphtheria toxoid [101]. This treatment modality resulted in a significantly improved PFS and OS when compared to controls. These promising results prompted a phase I trial to evaluate the feasibility and safety of pp65-DCs in combination with GM-CSF following host conditioning with dose intensified (DI)-TMZ. This trial was able to highlight that this treatment modality enhanced antigen-specific immunity and increased long-term PFS of 25.3 months and OS of 41.1 when compared to historical controls. Still, profound lymphopenia and increased Treg proportions following DI-TMZ were noted in patients. 

Vocimagene amiretrorepvec (Toca 511), a retroviral virus vector encoding cytosine deaminase, demonstrated a durable response rate of 21.7% in addition to a tolerable safety profile when compared to the standard of care in a newly diagnosed GBM [102]. In 2015, following this dose-escalation phase I trial, Toca 511 was investigated in the phase III Toca 5 trial (NCT02414165); however, this trial was later terminated [103]. ASPECT was a phase III trial that investigated sitimagene ceradenovac, an oncolytic vaccine comprised of replication-deficient human adenovirus type 5 that contains the HSV-TK gene as well as E1 and partial E3 deletions [104]. This trial randomized 250 patients to either standard therapy or standard therapy in addition to sitimagene ceradenovec. The primary endpoint was time to reintervention, defined as any kind of treatment (including chemotherapy, radiotherapy, or surgery) given to prolong survival following tumor progression, recurrence, or death. The time to recurrence was longer in the experimental arm, at 308 days, as compared to the control group, who had a time to reintervention of 268 days. Nevertheless, no difference in OS survival was observed between the two groups. More recent work has begun to explore alternative delivery strategies for oncolytic virus therapy in malignant gliomas. For example, Fares et al. investigated the delivery of an engineered oncolytic adenovirus via neural stem cells in their NSC-CRAd-S-pk7 vaccine. In addition to being able to provoke an immune response, encouraging survival outcomes were noted in their patient cohort, particularly with patients with MGMT-unmethylated tumors. 

### 2.5. Cytokine Therapy

Cytokines are secretory molecules that help facilitate communication for the innate and adaptive immune systems. Interferons (IFN) and interleukins (IL) represent two important types of cytokines that are capable of provoking antitumor effects [105,106]. IL-2 and IFN-α have been established as FDA-approved treatments for various hematological and nonhematological malignancies [107,108,109,110]. Therapies utilizing intratumoral injections of IL-2 genes in conjunction with herpes simplex virus type 1 thymidine kinase (HSV-TK) genes carried by retroviral vector-producing cells (RVPCs) have generated an antitumor response in select patients [111]. Other ILs have also been investigated. In another study, Okada et al. investigated the use of autologous fibroblasts containing HSV-TK and IL-4-encoding genes, when combined with irradiated autologous glioma cells, for the treatment of high-grade gliomas (grade III astrocytoma and GBMs). They were able to demonstrate a radiological and clinical improvement in two out of six patients who were able to complete the experimental regimen [112]. Weber et al. demonstrated a tolerable safety profile of their intratumorally injected vaccine consisting of IL-4 and Pseudomonas exotoxin (IL-4-PE) recombinant protein vaccine. In post-contrast MRI sequences, a decrease in signal intensity within the tumor was seen in patients following the administration of the vaccine; this is consistent with tumor necrosis [113].

IL-13 shares the same receptor as IL-4 and was used to create IL-13-PE38QQR, a recombinant protein vaccine composed of IL-13 and *Pseudomonas aeruginosa* exotoxin A [114]. Although the safety profile of this vaccine was confirmed in a phase I trial, a subsequent phase III study showed no significant difference in OS between the two groups [115]. Two phase II single-arm trials also highlighted the improved efficacy of TMZ with interferon-α in the treatment of recurrent GBM when compared to historical controls [116]. Interestingly, the radiological assessment showed that the use of cationic liposome-mediated interferon-beta (IFN-β) gene transfer treatment was able to induce a 50% tumor reduction in two out of five patients included in a pilot clinical trial [117]. Additionally, when compared to historical controls, this treatment lengthened the median OS. IFN-γ has also been investigated—however, improved efficacy of radiotherapy and chemoradiotherapy when combined with IFN-γ has not been noted for the treatment of GBM [118].

### 2.6. Chimeric Antigen Receptor (CAR) T-Cell Therapy

Chimeric antigen receptor (CAR) T-cell therapy utilizes specially engineered immune cells to target malignancies and is effective in treating certain hematologic malignancies (Figure 2). This treatment modality is unique in that it targets specific TAAs, which are commonly expressed on the surface of tumors and not on normal cells. EGFRvIII is a truncated receptor caused by mutations in the EGFR gene and serves as a common TAA in GBM. EGFRvIII is expressed in over one-third of newly diagnosed GBM patients and is not expressed on normal tissues, rendering it tumor-specific [119,120,121]. Johnson et al. showed that EGFRvIII-directed CAR T-cells were able to activate the immune system resulting in the regulation of tumor growth in xenogeneic subcutaneous and orthotopic models of human EGFRIII+ GBMv [122]. This study led to the creation of a phase I clinical trial to test the safety of EGFRvIII-directed CAR T-cells in patients with recurrent or residual GBM. This study was terminated early due to the sponsor’s decision to pursue combination therapies. In 2017, O’Rourke et al. conducted the first in-human study of EGFRvIII-directed autologous CAR T-cells delivered as a single intravenous dose [123]. This study included 10 patients with recurrent GBMs and sought to determine the safety and efficacy of this treatment modality. The results of this study were promising in that all patients exhibited detectable levels of EGFRvIII-directed CAR T-cells in peripheral blood. Additionally, seven patients received the surgical intervention after the CAR T-cell therapy, and tissue analysis showed in all seven patients. CAR T-cells were found in the region of active GBM. In five of the seven patients, there was a decrease in the EGFRvIII antigen. Nevertheless, when analyzing the tumor microenvironment, an increase in expression of Treg cells and inhibitory molecules were noted after treatment. Although CAR T-cell therapies showed promising initial results, overcoming the immunosuppressive tumor microenvironment remains a major challenge. 

Another widely investigated target is IL-13 receptor variants, as they have been previously shown to have increased expression in adult glioma cells [124,125]. Normally, activation of IL-13Rα1 has downstream effects, most notably the phosphorylation and activation of the STAT6 protein. A variant of the IL-13Rα1, IL-13Rα2, can be expressed in GBM, and this serves as a target TAA for CAR T-cell therapy. It is believed that the IL-13Rα2 receptor functions as a decoy receptor that binds IL-13 with higher affinity than IL-13Rα1 [126]. The subsequent alteration of this pathway contributes to the oncogenic nature of IL-13Rα2. Interestingly, IL13Rα2 expression alone stimulates GBM cells to invade surrounding structures, and the co-expression of IL13Rα2 in addition to EGFRvIII has been noted to confer GBM cell proliferation [127]. Studies of GBM have found IL-13Rα2 to be expressed in as many as 70% of analyzed cases [128,129].

The IL-13/STAT6 pathway is important in the transcription initiation of various downstream genes, including genes important for immune function [130,131]. Although there was an early promise for IL-13Rα2-directed CAR T-cell therapy, previous studies highlighted the limited persistence of these T-cells resulting in the recurrence of antigen-positive gliomas [132]. There is an open-label phase I trial (NCT04003649) evaluating the efficacy and safety of IL13Rα2-targeted CAR T-cells as a monotherapy as well as together with nivolumab and ipilimumab [133] in both recurrent and refractory GBMs. 

Human epidermal growth factor receptor 2 (HER2) is another TAA that is expressed in roughly 80% of GBMs [134,135]. The HER-2 receptor functions normally to activate various signal transduction pathways that ultimately allow for cell growth and differentiation [136]. Early work has highlighted the safety of HER2-specific CAR T-cell therapy and the potential clinical benefit for patients with progressive GBM. This was done through a phase I dose-escalation trial that evaluated the safety of HER2-specific CAR T-cells and their feasibility in augmenting antitumor activity in patients with GBM [137]. In addition to being well-tolerated, HER2-specific CAR T-cells were found in peripheral blood up to 12 months after administration. 

Hedge et al. aimed to establish the utility of a bispecific CAR molecule in improving the antitumor activity of T-cells [138]. The authors of the study targeted HER2 and IL13Rα2, two commonly found tumor antigens. A HER2-binding scFv and an IL13Rα2-binding IL-13 mutein were joined to make a tandem CAR exodomain (TanCAR) and a CD28.ζ endodomain. The TanCAR T-cells showed activation dynamics similar to those of monospecific CAR T-cells, as well as displaying antitumor capabilities. Interestingly, a super-additive effect on T-cell activation was observed, and authors attributed this to the induction of HER2-IL13Rα2 heterodimers by the TanCAR T-cells. Bispecific CAR T-cell therapy has been shown to enhance T-cell antitumor effects in addition to mitigating antigen escape. Still, patient variability of tumor surface antigens limits the use of therapies targeting 1 or 2 antigens. Bielamowicz et al. highlighted the feasibility of targeting three antigens using a single CAR T-cell product in their clinical trial [139]. Trivalent CAR T-cells targeting ephrin-A2 [EphA2], HER2, and IL13Rα2 were tested in 15 ex-vivo samples of primary GBM. The data from this study show that interpatient variability was overcome with the use of a trivalent CAR T-cell therapy in nearly all samples. Additionally, these CAR T-cells improved cytokine release and subsequently cytotoxicity when compared to monospecific or bispecific CAR T-cells in patient-derived xenograft models. Improved efficacy coupled with the ability to overcome variability among patient tumors highlights the potential utility of trivalent CAR T-cell therapy. 

In Sweden, a randomized phase II multicenter trial was conducted in 2015 to evaluate the efficacy and safety of immunotherapy called ALECSAT (Autologous Lymphoid Effector Cells Specific Against Tumor Cells) [140]. ALECSAT is an example of a form of adoptive cell transfer and uses ex-vivo activated autologous cytotoxic CD8+ T-cells as well as NK cells [141]. A pre-clinical trial was performed to determine the effect of ALECSAT on GBM cancer stem cells in vitro. This is of particular importance due to the role of cancer stem cells in the recurrence of GBM and their resistance to conventional therapy [142,143]. The results of this study helped to elucidate that, in fact, ALECSAT causes a robust cytotoxic dose-dependent response that preferentially targets GBM cancer stem cells. In addition to the cytotoxic response, a decrease in the proliferation of surviving cancer stem cells was also noted. Still, the optimal number of infused cells is an important parameter that must be established. 

Cytokine-induced killer (CIK) cells are a population of T-cells that have also been able to co-express natural killer (NK) cell surface molecules such as CD56 [144]. To create CIK cells, peripheral blood mononuclear cells are harvested and created in vitro through the addition of a CD3 monoclonal antibody (CD3mAb), IL-2, and IFN-γ [145]. CIK cells are a promising modality for the treatment of cancers due to their antitumor activity, as well as their high proliferation rate [146]. Additionally, they do not need to first recognize major histocompatibility complexes or antibodies to activate their cytotoxic activity. This allows for a potent and robust antitumor response with minimal cytotoxic effects on normal tissues. The efficacy of this immunotherapy has been validated in several clinical trials and has been demonstrated to be well-tolerated [147,148,149]. A multi-center, open-label phase III clinical trial conducted in South Korea aimed to evaluate the potential benefits of autologous CIK cells in the treatment of newly diagnosed GBM [150]. The CIK cell immunotherapy was evaluated in conjunction with the standard of care, while the control arm consisted of the standard of care alone. From December 2008 to October 2012, 180 patients were randomly assigned to the treatment or the control arm. The intention-to-treat analysis yielded a PFS of 8.1 months in the treatment arm versus 5.4 months in the control arm. Additionally, there was no significant difference between grade 3 or higher adverse events, performance status, or health-related quality of life between the two arms. Nevertheless, when comparing the two arms, there was no significant difference in OS.

## 3. Conclusions and Limitations

GBM presents unique challenges when approaching treatment from an immunotherapeutic angle. GBM is largely immunosuppressive to the systemic immune system as well as to neighboring cells alike [165]. This largely arises from the complex interactions of cytokines, extracellular matrix proteins, and other diverse cell populations. Although their complex interactions are not fully understood, studies have highlighted numerous examples of the far-reaching consequences of this microenvironment. Microglia and macrophages are examples of important players in the tumor microenvironment and adjacent regions. Along with myeloid-derived suppressor cells (MDSCs), these cells work in conjunction to inhibit cytotoxic T-cells as well as accentuate the effect of Treg cells [166,167,168]. More than just tumor cells influence immunosuppression; the tumor microenvironment is also a site of chronic inflammation. It is believed that these inflammatory stimuli serve to impact the blood-brain barrier (BBB) as well as activate microglia cells within the CNS [169]. These cells are recruited via various chemoattractants, and they can make up as much as 50% of the tumor mass [170]. 

One chemoattractant that has been studied is CX3CL1 which binds to the CX3CR1, a receptor found on a majority of microglia as well as some tumor-associated macrophages [171]. Once activated, the CX3CL1/CX3CR1 system is a source for the upregulated expression of gelatinases, such as MMP9, and the membrane-associated endopeptidase, MT1-MMP [172]. Multiple studies have proposed that microglial MMP9 enhances angiogenesis through VEGF regulation as well as promotes glioma motility [173,174,175]. Interestingly, tumor-associated macrophages are known to increase MMP9 expression via their release of TFG-β1 [176]. The release of TFG-β1 has also been noted to contribute to the enhanced invasive nature of glioma stem-like cells (GSLCs) [177]. 

Another important component of the tumor microenvironment is the endothelial cells of the vasculature surrounding the tumor. Through their interactions with the microenvironment, these cells contribute to poor outcomes seen in patients with GBM. In 2010, Ricci-Vitiani et al. showed that 20–90% of endothelial cells making up GBM-associated vasculature had the same mutations as the GBM cells themselves [178]. Another study was able to identify a population of CD133(+) tumor cells that express vascular-endothelial cadherin (CD144), signifying a majority of GBM-associated endothelial cells may arise from these specific tumors cells [179]. CD133(+) tumor cells have also been identified as a source of elevated expression of cytokines such as TGF-β and IL-10, which have been correlated with poorer outcomes [180,181]. Additionally, patients with recurrent grade III and IV gliomas following standard treatment were found to have tumors that contained a significantly larger amount of CD133+ cells [182,183]. Cytokines such as TGF-β and IL-10 are also secreted by T-reg cells and contribute to suppressing the antitumor response [184]. These cytokines also limit the amount of IFN-γ and IL-2 production, further contributing to the CD8+ T-cell anergy antitumor response seen [185,186,187]. In patients with GBM, an increased proportion of Tregs to CD4+ T-cells can be seen both at the location of the tumor as well as systemically [186,188].

T-cell sequestration is another unique challenge that arises in the setting of GBM. It has been shown that tumors in the CNS, specifically intracranial tumors, can cause extensive sequestration of immunocompetent cells in the bone marrow [189]. Furthermore, commonly used treatments such as temozolomide and high-dose corticosteroids also contribute to immunosuppression and lymphopenia [190,191]. Thus, there is no development of an effective antitumor response. The limited success of some immunotherapeutic strategies can also be accounted for by the T-cell exhaustion that is oftentimes attributed to the limited efficacy of checkpoint blockade treatment in GBM [192]. It is thought that continual interaction with antigens in suboptimal conditions contributes to T-cell exhaustion [193]. These T-cells go on to upregulate inhibitory immune checkpoints such as TIM3 and LAG3 [28]. The upregulation of these two immune checkpoints along with others such as 2B4, CD160, CD39, BTLA, and TIGIT have been shown, in other cancers, to be strongly associated with resistance to immune checkpoint blockade treatment [192]. 

Finally, another major challenge in treating GBM is the genetic heterogeneity these tumors display. This genetic heterogeneity can also be seen within individual tumors, which makes creating treatment plans based on a single biopsy extremely difficult [194,195,196]. Moreover, single-target therapies create selective pressures that can lead to antigen escape [197]. This phenomenon describes the expression of alternative forms of the original target antigen after initial treatment. Therefore, creating epitopes that cannot be recognized by CAR T-cell therapy is vital in maximizing therapeutic efficacy. Compared to other malignancies, GBM is also notable for a relatively low mutational burden [198]. It has been shown that GBM tumors contain only 30 to 50 non-synonymous mutations [199]. Usually, proteins encoded from mutated genes within tumors yield antigens that are exclusive to the tumor. A small percentage of these tumor-specific antigens (TSAs) are processed into neoepitopes, which are subsequently presented to T-cells eliciting a response [200]. TAAs are antigens that can be found expressed by both tumor cells and normal cells. Attempting to target these antigens can result in collateral autoimmunity, and side effects such as encephalitis have been demonstrated in animal models [201]. This highlights the difficulty in creating vaccine-based strategies due to the relative lack of TSAs. 

## Figures and Tables

**Figure 1 cancers-13-04548-f001:**
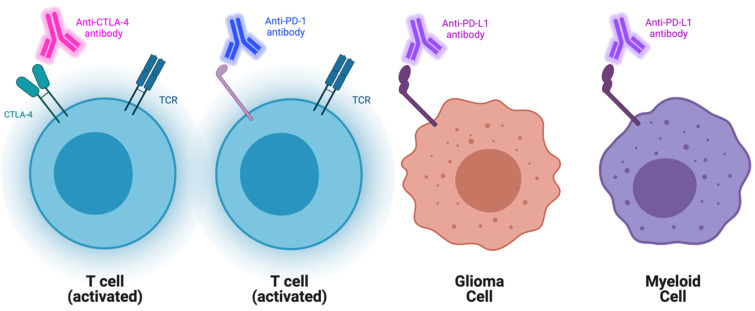
Schematic representation of current checkpoint blockades and their targets. Represented aCTLA-4, PD1, and PDL1 blockades and their main cellular targets.

**Figure 2 cancers-13-04548-f002:**
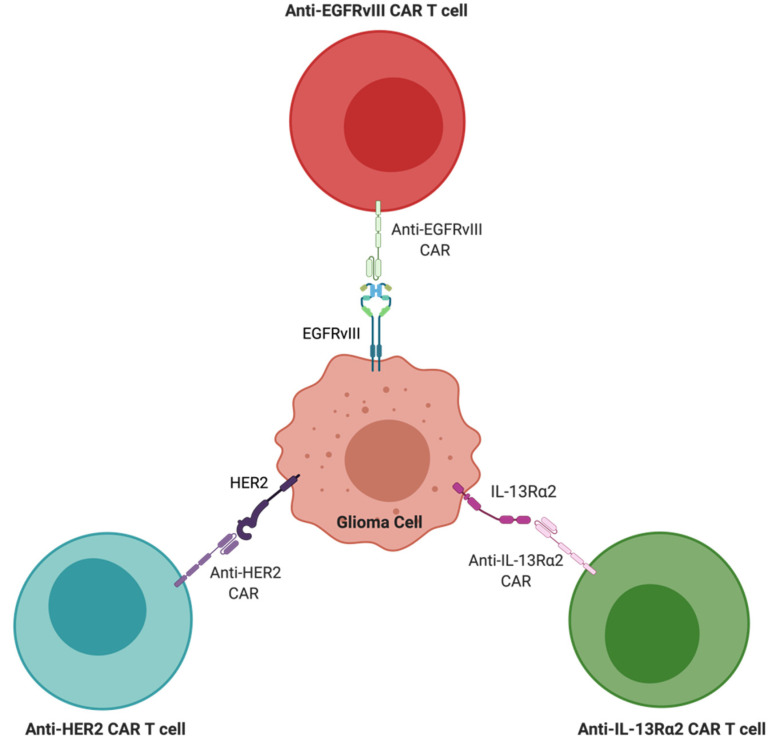
Schematic representation of different CAR T-cells and their targets. Represented anti-HER2, anti-IL13Rα2, and anti-EGFRvIII CARs. All three antigens are represented in the same cell for the purpose of the graphic.

**Table 1 cancers-13-04548-t001:** Current immunotherapeutic strategies used in clinical trials.

Class	Target	Intervention	Comments	References
Checkpoint Inhibitor	PD-1 & CTLA-4	Treatment Arm 1: nivolumab Treatment Arm 2: nivolumab + ipilimumab	Overall survival (OS) at 6 months was 75% among the 20 treated patients; promising compared to historical controls	Omuro, A; Vlahovic, G; Lim, M; et al. [28]
Checkpoint Inhibitor	PD-1	Nivolumab + RT	Discontinued due to inability to meet primary endpoint	Bristol Meyer Squibb press release [151]
Checkpoint Inhibitor	PD-1	Nivolumab	No statistically significant improvement in PFS noted	Bristol Meyer Squibb press release [152]
Checkpoint Inhibitor	PD-1	Neoadjuvant Nivolumab	Increased immune cell infiltration and T-cell receptor clonal diversity; no clear benefit shown following salvage surgery	Schalper, K.A.; Rodriguez-Ruiz, M.E.; Diez-Valle, R.; et al. [38]
Checkpoint Inhibitor	PD-1	Treatment Arm 1: Neoadjuvant pembrolizumab with continued adjuvant therapy following surgery Treatment Arm 2: Adjuvant pembrolizumab following surgery only	Prolonged overall survival was found to be statically significant in the neoadjuvant group	Cloughesy, T.F.; Mochizuki, A.Y.; Orpilla, J.R. et al. [39]
Checkpoint Inhibitor	PD-L1	Durvalumab in addition to the radiotherapy (60 Grays over 30 fractions) followed by durva monotherapy	Median OS was 15.1 months with OS of 12 months seen in 60% of patients from this study	Reardon, D; Kaley, T; Dietrich, J; et al. [153]
Checkpoint Inhibitor	CTLA-4	Adjuvant Ipilimumab	Ongoing	Clinical Trial NCT03460782 [154]
Checkpoint Inhibitor	IDO	Indoximod + standard of care	Ongoing	Clinical Trial NCT02052648 [155]
Checkpoint Inhibitor	PD-1 & CTLA-4	Treatment Arm 1: Ipilimumab Treatment Arm 2: Nivolumab Treatment Arm 3: Ipilimumab + Nivolumab	Ongoing	Clinical Trial NCT02311920 [156]
Checkpoint Inhibitor	PD1 & CTLA-4	Treatment Arm 1: Adjuvant trememlimumab Treatment Arm 2: Adjuvant durvalumab Treatment Arm 3: Adjuvant trememlimumab + durvalumab	Ongoing	Clinical Trial NCT02794883 [34]
Checkpoint Inhibitor	PD-1 & anti-CD27	Varlilumab + nivolumab	Ongoing	Clinical Trial NCT02335918 [157]
Checkpoint Inhibitor	IDO-1	Epacadostat + Nivolumab	Ongoing	Clinical Trial NCT02327078 [46]
Checkpoint Inhibitor	Anti-LAG-3 & anti-CD137 & PD-1	Treatment Arm 1: BMS-986016 Treatment Arm 2: Urelumab Treatment Arm 3: BMS-986016 + Nivolumab Treatment Arm 4: Urelumab + Nivolumab	Ongoing	Clinical Trial NCT02658981 [158]
Peptide-based Vaccine	EGFRvIII	ACTIVATE: Investigated use of rindopepimut alone with standard of care ACT II and ACT III: Investigated use of rindopepimut with adjuvant TMZ ACT IV: rindopepimut + standard of care in patients with minimal residual disease	ACT II and ACT III: PFS of roughly 15 months from time of diagnosis and OS of 24 months compared to cohort that received standard of care ACT IV: No significant difference in OS	Sampson, J.H.; Heimberger, A.B.; Archer, G.E.; et al. [47] Sampson, J.H.; A, K.D.; Archer, G.E.; et al. [48] Schuster, J; Lai, R.K.; Recht, L.D.; et al. [49] Weller, M; Butowski, N; Tran, D.D.; et al. [50]
Peptide-based Vaccine	Tumor associated antigens	Treatment Arm 1: ICT-107 Treatment Arm 2: Standard of care	Trial suspended due to lack of funding	Clinical Trial NCT02546102 [84]
Peptide-based Vaccine	IDH1	Safety and feasibility trial	Ongoing	Clinical Trial NCT02454634 [61] Clinical trial NCT02193347 [62]
Peptide-based Vaccine	Neoantigen vaccine (APVAC1 & APVAC2)	Safety and feasibility trial	The 15 vaccinated patients had a median OS of 29 months	Keskin, D.B.; Anandappa, A.J.; Sun, J.; et al. [74]
Peptide-based Vaccine	Heat Shocked Proteins	HSPPC-96 vaccine Treatment Arm 1: HSPPC-96 with concomitant bevacizumab Treatment Arm 2: HSPPC-96 with administration of bevacizumab at tumor progression	Ongoing	Clinical Trial NCT01814813 [159]
Peptide-based Vaccine	Tumor Associated Antigens: Antineoplastons	Treatment Arm 1: Radiation and chemotherapy alone Treatment Arm 2: antineoplaston	In the RGBM population, 6.7% of patients had complete responses; 6.7% had partial responses, with 16.7% PFS seen at 6 months Overall survival for the RGBM group was 34.7% at one year and 3.47% at 2, 5, and 10 years In the ERGBM cohort, complete responses occurred in 8.3% of patients; partial response occurred in 8.3% of patients Progression-free survival at 6 months was 20.8%, overall survival was 39.3% at 1 year and 4.4% at 2, 5, and 10 years	Burzynski, S; Janicki, T; and Burzynski, G. [160]
Peptide-based Vaccine	APVAC1 & 2: which targeted (HLA)-A*02:01 or HLA-A*24:02 neoantigens	APVAC1&2 vaccines	APVAC1 vaccines were able to elicit sustained CD8+ T-cell response, while the APVAC2 elicited a CD4+ T-cell response; both vaccines were well tolerated	Hilf, N.; Kuttruff-Coqui, S.; Frenzel, K.; et al. [73]
Peptide-based Vaccine	CMV protein-pp65 antigen	CMV pp65-specific dendritic cells (pp65-DCs) when combined with vaccine site pre-conditioning using tetanus-diphtheria toxoid	Highlighted that this treatment modality enhanced antigen-specific immunity and increased long-term PFS and OS Profound lymphopenia and increased Treg proportions following DI-TMX	Batich, K.A.; Reap, E.A.; Archer, G.E.; et al. [101]
Peptide-based Vaccine	Ad-RTS-hiL-12	Ad-RTS-hiL-12 is a non-pathogenic form of an adenovirus that has been genetically modified to encode the IL-12 protein; Veledimex serves as an oral ligand activator for IL-12.	Ongoing	Clinical Trial NCT03636477 [161]
Cell-based Vaccine	Autologous glioma cells	Autologous glioma cells mixed with irradiated GM-K562 cells	T- lymphocyte activation with significant increased expression of PD-1 and 4-1BB by CD8+ cells and CTLA-4, PD-1, 4-1BB, and OX40 by CD4+ cells Vaccination resulted in increased frequency of regulatory CD4+ T-lymphocytes	Curry, W; Gorrepati, R; Piesche, M; et al. [162]
Cell-based Vaccine	Autologous glioma cells	DCV vaccine	Statistically significant median overall survival seen in treatment group (480 vs. 400 days)	Yamanaka, R.; Homma, J.; Yajima, N.; et al. [79]
Cell-based Vaccine	Autologous glioma cells	Gliovax vaccine	Ongoing	Rapp, M.; Grauer, O.M.; Kamp, M.; et al. [85]
Cell-based Vaccine	Autologous glioma cells	DCVax^®^-L vaccine	Median OS (mOS) for the intent-to-treat (ITT) population was 23.1 months from surgery; an improvement from the typical mOS for the standard of care	Patente, T.A.; Pinho, M.P.; Oliveira, A.A.; et al. [75]
Cell-based Vaccine	Autologous glioma cells Target: EGFRvIII		EGFRvIII-directed CAR T-cells were able to activate the immune system resulting in regulation of tumor growth in xenogeneic subcutaneous and orthotopic models of human EGFRIII + GBM	Johnson, L.A.; Scholler, J; Ohkuri, T; et al. [122]
Cell-based Vaccine	Autologous glioma cells Target: IL13Rα2	Vaccine	Anti-glioma responses observed in 2 patients, 1 had increased tumor necrotic volume on MRI at administration site	Brown, C; Badie, B; Barish, M; et al. [163]
Cell-based Vaccine	Autologous glioma cells Target: IL13Rα2	Vaccine	Ongoing	Clinical Trial NCT004003649 [133]
Cell-based Vaccine	Autologous glioma cells Target: IL13Rα2	Vaccine	IL13Rα2-CAR.IL15 T-cells in vivo had a greater persistence and anti-tumor activity than IL13Rα2-CAR T-cells	Krenciute, G; Prinzing, B; Yi, Z; et al. [164]
Cell-based Vaccine	Autologous glioma cells Targets: IL13Rα2 & HER2	Vaccine	Super-additive effect on T-cell activation	Hegde, M.; Mukherjee, M.; Grada, Z.; et al. [138]
Cell-based Vaccine	Autologous glioma cells Targets: IL13Rα2 & HER2 & EphA2	Vaccine	Improved survival of treated animals highlight the utility of trivalent CAR T-cell therapy	Bielamowicz, K.; Fousek, K.; Byrd, T.T.; et al. [139]
Cell-based Vaccine	Autologous glioma cells	Vaccine	ALECSAT exhibited a robust cytotoxic dose-dependent response that preferentially targets GBM CSCs Decreased proliferation of surviving CSC Number of infused cells is important and must be established.	Wenger, A.; Werlenius, K.; Hallner, A.; et al. [141]
Cell-based Vaccine	Autologous glioma cells	Vaccine	Intention-to-treat analysis yielded a PFS of 8.1 months in the treatment arm versus 5.4 months in the control arm	Kong, D.S.; Nam, D.H.; Kang, S.H.; et al. [150]
Cell-based Vaccine	Autologous glioma cells Target: EGFRvIII	Vaccine	In 7 patients, CAR T-cells were found in the region of active GBM; 5/7 seven patients had decreased EGFRvIII antigens	O’Rourke, D.M.; Nasrallah, M.P.; Desai, A.; et al. [123]

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
