# Peer review of "Current Immunotherapeutic Strategies for the Treatment of Glioblastoma"

_cancers, 2021, doi:10.3390/cancers13184548_

Round 1

Reviewer 1 Report

Well-organized, great review that I enjoyed reading. The authors put an extensive amount of work into it. I believe it will be well received by the experts in the field. I only have one stylistic suggestion regarding Table 1. I recommend making the first column a bit bigger so it doesn't cut the words in the middle, and narrowing down the second column. 

Author Response

We would like to thank you for your time, dedication and constructive criticism which we believe contributed to a more comprehensive revised manuscript. We have addressed all the raised questions and hope this new version of the manuscript fulfills the reviewers’ expectations.

We took the reviewer’s advice and made changes to the table to better accomadate the text.

Reviewer 2 Report

In this article, the authors describe the current immunotherapeutic strategies for the treatment of glioblastoma based on the results of clinical trials. The structure of the article is nice, the treatments and results are appropriately described, and in general the manuscript is well written. The figures and table are good and appropriate. I believe this article represents a good overview of immunotherapeutic approaches for glioblastoma, which can be useful for clinicians, scientists, and a general readership. I have only some minor comments, which I hope could help improve the quality of this manuscript.

  1. “TMZ (75 mg/m2) with concomitant RT” It would be good to spell out the full name of TMZ and RT at first occurrence.
  2. The first figure caption is reported as “Figure 2”.
  3. “This is particularly deleterious on the immune system when because temozolomide and high-dose corticosteroids commonly used in the treatment of GBM led to immunosuppression as well as lymphopenia” Please review this sentence because there’s probably a typo or a grammar mistake.
  4. In table 1, it would be great to report the references either in a standard and consistent format throughout the table, or with the numbers corresponding to the reference number in the reference list.

Author Response

We would like to thank you for your time, dedication and constructive criticism which we believe contributed to a more comprehensive revised manuscript. We have addressed all the raised questions and hope this new version of the manuscript fulfills the reviewers’ expectations.

1.“TMZ (75 mg/m2) with concomitant RT” It would be good to spell out the full name of TMZ and RT at first occurrence.

The sentence was changed to “The Stupp protocol entails the administration of temozolomide (TMZ) (75 mg/m2 ) with concomitant radiotherapy (RT) (60 Gy) for 42 days followed by 6 cycles of adjuvant TMZ (150 to 200 mg/m2) administered for 5 days during each 28-day cycle[8]” and this change was highlighted in blue.

2.The first figure caption is reported as “Figure 2”.

The first caption was relabled to correctly read “Figure 1”.

 3.“This is particularly deleterious on the immune system when because temozolomide and high-dose corticosteroids commonly used in the treatment of GBM led to immunosuppression as well as lymphopenia” Please review this sentence because there’s probably a typo or a grammar mistake.

We changed the sentence to now read “Furthermore, commonly used treatments such as temozolomide as well as high-dose corticosteroids also contribute to immunosuppression and lymphopenia”. We also highlighted the changed text within the document in blue.

4.In table 1, it would be great to report the references either in a standard and consistent format throughout the table, or with the numbers corresponding to the reference number in the reference list.

The References section was changed to cite sources in a consistent format that is easier for readers.